# Characterization of the Mitochondrial Genome of the Vietnamese Central Highland Wild Boar (*Sus scrofa*)

**DOI:** 10.3390/ani15142029

**Published:** 2025-07-10

**Authors:** Minh Thi Tran, Anh Le Hong Vo, Chi Nguyen Quynh Ho, Manh Quang Vu, Quan Minh To, Mai Thi Phuong Nguyen, Loan Thi Tung Dang, Nhan Lu Chinh Phan, Chung Chinh Doan, Huy Nghia Quang Hoang, Cuong Phan Minh Le, Son Nghia Hoang, Han Thai Minh Nguyen, Long Thanh Le

**Affiliations:** 1Faculty of Applied Technology, Van Lang University, 69/68 Dang Thuy Tram Street, Binh Loi Trung Ward, Ho Chi Minh City 70000, Vietnam; minh.tt@vlu.edu.vn; 2Faculty of Biology and Biotechnology, University of Science, Vietnam National University, Ho Chi Minh City 70000, Vietnam; honganh99826@gmail.com (A.L.H.V.); tomquan@hcmus.edu.vn (Q.M.T.); dttloan@hcmus.edu.vn (L.T.T.D.); 3Institute of Life Sciences, Vietnam Academy of Science and Technology, Ho Chi Minh City 70000, Vietnam; hnqchi@ils.vast.vn (C.N.Q.H.); ntpmai@tni.vast.vn (M.T.P.N.); plcnhan@ils.vast.vn (N.L.C.P.); doanchinhchung@gmail.com (C.C.D.); hnqhuy@ils.vast.vn (H.N.Q.H.); lpmcuong@ils.vast.vn (C.P.M.L.); hoangnghiason@yahoo.com (S.N.H.); han.nguyen@unh.edu (H.T.M.N.); 4Institute of Life Science and Application, Hoa Binh University (HBU), Nam Tu Liem, Hanoi 100000, Vietnam; vuqmanh@gmail.com; 5Center for Biotechnology and Genomic Medicine, Medical College of Georgia, Augusta University, Augusta, GA 30912, USA; 6Life Science Department, University of New Hampshire at Manchester, Manchester, NH 03101, USA

**Keywords:** Vietnamese pig, complete mitogenome, phylogenetic

## Abstract

Vietnamese Central Highland wild boars are under serious threat due to illegal poaching, habitat destruction, and uncontrolled breeding with domestic pigs. These issues can damage the unique genetic traits of wild boars, making conservation more difficult. At present, very little is known about the genetics of these animals. This study characterizes the complete genetic material (mtDNA) found in the mitochondria—a small part of the cell responsible for energy production—of the Vietnamese Central Highland wild boar. The molecule is 16,581 base pairs long and contains 37 genes with important functions. We identified unique features in one mitochondrial gene, which help trace the maternal origins of Vietnamese Central Highland wild boar populations. Our findings suggest that Vietnamese wild boars are more closely related to Asian pigs. This study adds value to current knowledge on the genetics of Vietnamese wild boars, contributing to future protection, conservation, and possibly breeding of these animals to prevent further decline.

## 1. Introduction

Located in the Indo-Burma biodiversity hotspot, which is well known for its largest wild boar diversity in the world, Vietnam is home to two wild boar species: *Sus scrofa* and *Sus bucculentus* [1]. While *Sus bucculentus* was claimed extinct, *Sus scrofa* has a rather wide distribution across the country, inhabiting wetlands and forests from the North to the Central Highlands [2,3]. Vietnamese Central Highland wild boars can be distinguished from domestic pigs through distinct phenotypic traits such as a small head, long snout, and a compact body shape coated with black or gray fur [2]. Wild boars exhibit remarkable intrinsic properties, ranging from survival resilience, adaptive capabilities, and disease resistance to good-quality meat and a high reproduction rate. Thus, wild boar farming holds significant importance for agricultural development in rural communities. Hybridization between wild boars, especially those of unknown origins, and domestic pig breeds poses a serious threat to the genetic integrity of natural wild boar populations [4,5,6,7]. This consequently leads to mixed genotypes and loss of unique traits, putting the number of genetically pure Vietnamese wild boars on rapid decline and ultimately altering wild boar genetic resources [8]. Until now, the genetic data of Vietnamese wild boars still remain underexplored.

The mitochondrial DNA (mtDNA) is a circular, double-stranded DNA molecule, containing 37 genes and a non-coding control region (D-loop). Mammalian mtDNA is approximately 16.5 kb in size and is characterized by high copy number, maternal inheritance, haploid nature, and absence of recombination [9]. Such characteristics make mtDNA a valuable molecular marker, providing insights into the study of genetic diversity, population history, and evolutionary origins [10]. Of the mitochondrial genome, the *cytochrome b* gene has been extensively studied to assess the genetic relationship among various pig breeds, serving as an essential tool for phylogenetic analysis and species identification [11,12,13,14]. An evaluation of mtDNA showed that native Vietnamese pigs were genetically diverse, sharing genetic patterns similar to pigs originating from neighboring countries [15,16]. Another study on the sequence variability of *cytochrome b* among Vietnamese Central Highland pigs revealed the close relationship between Vietnamese wild boars and Asian pig breeds [17]. Recently, the complete mitochondrial genome of several Vietnamese domestic pig breeds, namely I pig, Dong Khe pig, and Ha Lang pig, has been reported, aiding in the efforts to trace the origin of Vietnamese pig populations [18,19,20]. However, the mitochondrial genome of the Vietnamese wild boar and its phylogenetic background are still poorly understood.

This study reports the complete mitochondrial genome of the Vietnamese Central Highland wild boar and compares it with other wild boars and domestic pig breeds. We focus on characterizing mtDNA features and investigating the sequence variability of whole mitogenome sequences, providing valuable information for conservation and genetic research of Vietnamese wild boars.

## 2. Materials and Methods

### 2.1. DNA Extraction and Sequencing

A testicular tissue sample of the Vietnamese Central Highland *Sus scrofa* was provided by the Biological Museum, Tay Nguyen Institute for Scientific Research, Vietnam Academy of Science and Technology. The sample was stored at 4 °C in ethanol until DNA extraction. Total genomic DNA was extracted using the QIAmp DNA Mini Kit (Qiagen, Germantown, MD, USA) following the manufacturer’s instructions. The integrity of extracted DNA was checked on agarose gel electrophoresis. DNA quantification and purity determination were performed using a Qubit spectrofluorometer (Thermo Fisher Scientific Inc., Waltham, MA, USA). Samples having a concentration ≥ 2 ng/μL with a total yielded amount ≥ 90 ng and an OD260/OD208 ratio ≥ 1.70 were considered suitable for further experiments. Samples with DNA size <1000 bp were flagged.

Whole-genome sequencing libraries were prepared using the NEBNext Ultra II DNA Library Prep Kit for the Illumina platform (New England Biolabs, Ipswich, MA, USA). The library concentration was measured fluorometrically, and the average library size was determined using a bioanalyzer (Agilent, Santa Clara, CA, USA) according to Illumina’s library evaluation guidelines. Samples with concentration ≥ 0.50 ng/μL (for genome size < 1 Gb) or ≥2 ng/μL (for genome size > 1 Gb) were considered suitable for sequencing. Sequencing was subsequently performed using an Illumina sequencing system (Illumina, San Diego, CA, USA).

### 2.2. De Novo Assembly

Fragments of various sizes were assembled using the GetOrganelle (v1.7.7.0) pipeline with optimal parameters to construct a complete mitochondrial genome without the use of a reference genome. De novo assembly results are presented in Appendix A, showing a total contig length of 16,581 bp. This length is consistent with the size of vertebrate mitochondrial genomes, which typically range from 14 to 20 kb [21].

### 2.3. Bioinformatics Analysis

The raw sequencing data were preprocessed using fastp v0.23.1 and stored in FASTQ file format (Supplementary Data S1) [22]. Based on the Phred score value recorded for each nucleotide, unreliable or unidentified nucleotides with poor sequencing quality (type N nucleotides) were eliminated. The reading quality results are shown in Appendix A. The de novo assembly of purified reads was conducted using GetOrganelle v1.7.7.0 software [23]. Assembly quality was evaluated using Quast v5.2.0 [24] and by locally aligning reads to the assembled contigs. This enables the detection of regions with unusually low values of depth coverage compared to neighboring regions, which were noted in the assembly results. The annotation of the assembled mitogenome was completed using Mito FISH version 4.03 [25] and the MITOS2 tool in the Galaxy webserver platform [26], with a specialized database for mitochondrial genomes. The mitogenome sequence of the Vietnamese Central Highland *Sus scrofa* was deposited in GenBank with an accession number PV693689.

The graphical genetic map of the circular *Sus scrofa* mitogenome was constructed using Mito Annotator version 4.03 [25]. The amino acid and nucleotide compositions of the entire mitogenome were determined and compared with those of other *Sus scrofa* using MEGA12 [27]. Strand asymmetry was calculated using the following formulas: AT skew = (A − T)/(A + T) and GC skew = (G − C)/(G + C) [28]. Specific regions of transfer RNA (tRNA), PCGs, ribosomal RNA (rRNA), or control were determined by aligning the *Sus scrofa* mitogenome with the homologues of similar species. The secondary structures of 22 tRNAs were predicted using tRNAscan-SE software v2.0 [29] and Mitos Webserver [26]. The Relative Synonymous Codon Usage (RSCU) values of *Sus scrofa* mitogenome were computed using MEGA12 [27].

### 2.4. Phylogenetic Tree

To determine the molecular location of the Vietnamese Central Highland *Sus scrofa* in the evolutionary tree and its genetic relationship with other pigs, a total of 32 *Sus scrofa* whole mitogenomes were subjected to phylogenetic analysis. This includes domestic pig breeds from Asia (8), Europe (4), and Vietnam (4), and wild boars from Europe (5), China (4), Korea (2), Vietnam (2), India (2) and Malaysia, with *Phacochoerus africanus* used as an outgroup [30]. The mitogenomes were aligned using the ClustalW algorithm of MEGA12 [27]. The Tamura and Nei model was used as a genetic distance model [31]. A Neighbor-Joining tree was constructed using 1000 bootstrap replicates.

## 3. Results

### 3.1. Complete Mitochondrial Genome Analysis

The complete mitochondrial genome of the Vietnamese Central Highland *Sus scrofa* is 16,581 bp in length, which is smaller than the previously published mitogenomes of *Sus scrofa* from other regions, such as Europe (KP301137, 16,770 bp), China (EF545573, 16,620 bp), and India (MG725630, 16,738 bp). It encodes a total of 37 genes, including 13 protein-coding genes (PCGs), 22 transfer RNA (tRNA) genes, 2 ribosomal RNA (rRNA) genes, and a non-coding control region, as typically reported in vertebrates. Apart from *nad6* and 8 tRNA genes (tRNA-*Gln*, tRNA-*Ala*, tRNA-*Asn*, tRNA-*Cys*, tRNA-*Tyr*, tRNA-*Ser*, tRNA-*Glu*, and tRNA-*Pro*), which were located on the light (L) strand, the remaining 28 genes of the mitogenome were found on the heavy (H) strand (Figure 1 and Table 1).

### 3.2. Nucleotide Composition Pattern

The whole Vietnamese *Sus scrofa* mitogenome shows a similarly high A + T content to other *Sus scrofa*, accounting for 60.6% (Table 2), with the highest values being recorded in *trnF* (78.3%) and *trnH* (76.8%) [32]. mtDNA demonstrates a noticeable bias towards A/T in codon usage, with the third nucleotide in most codons tending to be an A or T base (Figure 2). A significantly similar pattern of nucleotide composition was identified among 15 *Sus scrofa* from different geographic regions and 2 members of the Suidae family. The length and A + T content did not vary significantly despite their diverse habitats, highlighting the conserved nature in the mitogenome of *Sus scrofa*. Of these, the amino acid distribution among the wild boars was significantly similar (Figure 3). The most frequently observed amino acids included Leu (15.23–15.28%), Ile (8.85–8.9%), Thr (8.2–8.4%), Ser (7.4–7.5%), and Met (6.9–7.0%), whereas Cys was rare (0.66–0.69%).

### 3.3. Protein-Coding Genes

Consistent with other vertebrates, the *Sus scrofa* mitochondrial genome also contains 13 core PCGs. The concatenated PCG sequence was estimated to span 11,342 bp, constituting 68.4% of the complete mitogenome. Of the 13 PCGs, 12 PCGs were located on the H strand (majority strand), whereas *nad6* was located on the L strand (minority strand). The overall A + T content of the PCGs was 60.3%, ranging from 56.1% (*cox3*) to 67.6% (*atp8*), which is consistent with the nucleotide composition pattern of the whole mitogenome.

The strand asymmetry of the *Sus scrofa* mitogenome across the PCGs was assessed through AT and GC skews. All the PCGs exhibited a positive AT skew and a negative GC skew, reflecting a particular preference towards adenine and cytosine (Table 3). This pattern is consistently found in *Sus scrofa* across other regions, where GC skew commonly ranged from −0.402 to −0.356. Among the PCGs, *nad6* displayed the highest AT skew (0.349) and the lowest GC skew (−0.562). Interestingly, the adenine and thymine contents in *cox1* were nearly equal, represented through an AT skew of 0.001 (Figure 4).

Additionally, all PCGs in *Sus scrofa* began with the start codon ATN (ATG or ATT), except for *nad4l*, which started with GTG. Incomplete termination codons (T--) were observed in six PCGs, namely *nad1*, *nad2*, *cox2*, *cox3*, *nad3*, and *nad4*.

### 3.4. Control Region

The control region (D-loop) is rich in A + T content and located between the tRNA-*Pro* and tRNA-*Phe* genes. It is 1145 bp in length (positions 15,781–16,581 and continuing at 1–344), typical for vertebrate mitogenomes [33]. An alignment of 20 *Sus scrofa* control regions revealed an 11 bp repeat consensus (CGTACACGTG) starting at site 705. The Vietnamese Central Highland wild boar harbors only 9 copies of this repeat, while up to 45 copies were found in the European wild boar (FJ237000). In addition, the A + T content and AT and GC skews of the control region were 59.7%, −0.15, and −0.29, respectively. Interestingly, the D-loop of the Vietnamese Central Highland *Sus scrofa* also displayed the highest adenine and cytosine bias compared to 19 other pigs.

Further investigation detected 29 polymorphic sites among 20 *Sus scrofa* samples within the control region, most of which were similarly found in different *Sus scrofa* individuals (Appendix A).

### 3.5. tRNA and rRNA Genes

Notably, 22 genes encoding 22 tRNAs were identified by tRNAscan-SE, each corresponding to a specific amino acid, while serine and leucine were each transferred by two different tRNAs. The predicted tRNA genes could be folded into a typical secondary clover-leaf structure, except for the D-arm lacking *trnS1*, which could not form a stable structure (Figure 5). The gene length varied for each tRNA, ranging from 59 bp (*trnS1*) to 75 bp (*trnL2* and *trnS2*). Of the 22 tRNA genes, 8 were located on the L strand, while the remaining were located on the H strand. The average base composition of the tRNA genes was A: 32.2%, T: 30.6%, G: 19.7%, and C: 17.5%, with the highest and lowest GC content observed in *trnM* (47.1%) and *trnR* (21.7%), respectively. In addition, a total of seven mismatches were detected in six tRNA genes, located in the amino acid acceptor (AA) stem, the pseudouridine (TΨC) stem, and the anticodon (AC) stem (Table 4). The structure and number of tRNA genes found in this study are similar to previous reports on *Sus scrofa* [18,32].

Two rRNA genes encoding for the small ribosomal subunit (12 S) and large ribosomal subunit (16 S) were positioned between *trnF* and *trnV* genes, and between *trnV* and *trnL2* genes, respectively. The lengths for the 12 S and 16 S rRNAs were 962 bp and 1572 bp, respectively.

### 3.6. Overlapping and Intergenic Regions

The Vietnamese Central Highland *Sus scrofa* mitogenome had nine overlapping sequences among all the genes, with a total length of 76 bp, ranging from 1 bp to 43 bp in length (Table 1). The overlaps were located on both the H strand and L strand, four of which resided within the PCGs, and the remaining were observed between the tRNA and rRNA genes. Notably, the overlap between the *trnV* and *rrnL* (2 bp) was uniquely identified in the Vietnamese Central Highland *Sus scrofa.* The 1 bp overlap between *trnV* and *rrnS* (1 bp) was also found in I pig [18], whereas the other overlapping regions are commonly present in other *Sus scrofa* mitogenomes, with the longest observed between *atp8* and *atp6* (43 bp).

There are 10 intergenic spaces among all genes, ranging from 1 bp to 32 bp and making up a total of 58 bp (Table 1). The longest space (32 bp) was located between *trnN* and *trnC*, as typically observed in other wild boars and pig breeds.

### 3.7. Mutations

A comparative analysis of 13-PCG concatenated sequences of Vietnamese Central Highland wild boar and 19 other *Sus scrofa* revealed 232 polymorphic sites, resulting in 51 amino acid substitutions (Appendix A). The Vietnamese Central Highland *Sus scrofa* and Asian *Sus scrofa* shared an almost similar amino acid profile, while some variations were only present in European *Sus scrofa*. No amino acid alteration was unique to the Vietnamese Central Highland *Sus scrofa*, although three mutations at 1461, 4577, and 8367 positions were found only in the Vietnamese Central Highland wild boar, Dong Khe pig breed, and Ha Lang pig breed, which are of Vietnamese origin. The Vietnamese Central Highland wild boar differed from the Vietnamese wild boar in 13 nucleotides, resulting in 2 amino acid changes.

Further evaluation in partial *cytochrome b* sequences of 8 Vietnamese wild boars (7 of which were collected from the Central Highlands) revealed 15 variable positions, representing 3.11% of the total length of the DNA sequences analyzed (Appendix A). 

### 3.8. Phylogenetic Analysis

The genetic relationship of the Vietnamese Central Highland wild boar with 31 other wild boars and domestic pig breeds is depicted in the neighbor-joining phylogenetic tree (Figure 6). The tree topology was similar to previous studies in which pigs formed two distinct monophyletic clades corresponding to their Asian and European origins, supported by a bootstrap value of 98% and 89%, respectively. Indian wild boars and Malaysian wild boars fell out of these clades. The two Vietnamese wild boars belonged to two different groups within the Asian clade. The Vietnamese Central Highland wild boar formed a subgroup with Dong Khe pig and Ha Lang pig with a 96% bootstrap support, while the Vietnamese wild boar clustered with the I pig.

## 4. Discussion

This study reports the molecular characterization of the complete mitochondrial genome of Vietnamese Central Highland wild boars, which comprises 37 genes encoding for 2 RNA subunits, 22 tRNAs, and 13 mitochondrial proteins. The mitogenome was comparable with earlier reports of other wild boars and pig breeds in various perspectives, highlighting the conserved nature and important function of mtDNA. The genetic diversity of the Vietnamese Central Highland *Sus scrofa* compared with other wild boars and domestic pig breeds’ taxa was reflected through differences in their structural and nucleotide composition patterns. These variations were analyzed based on specific gene sequences, AT content, and skewness indices.

The Vietnamese Central Highland wild boar exhibited incomplete termination codons in 6 out of the 13 PCGs, as similarly observed in Chinese (EF545573, EU333163, EF545569, and EF545572) and Indian wild boars (MG725630, MG725631, and OM162160). In contrast, only four PCGs (*nad1*, *nad2*, *cox2*, and *nad3*) in European (FJ237000, FJ237001, FJ237002, and FJ237003) and Korean (AY574047 and DQ268530) wild boars lack a complete termination codon. Incomplete termination codons can be post-transcriptionally completed by polyadenylation of mRNA, a common feature in mammalian mitochondrial genomes [34]. This variation may reflect regional differences in mitochondrial genome evolution, suggesting a more conserved mitochondrial structure or shared evolutionary ancestry in different wild boars of Asian lineages compared to European wild boars.

The control region of Vietnamese Central Highland *Sus scrofa* is relatively small compared to what was observed in other pigs across the world, such as European wild boars (FJ237000, 1505 bp), Korean wild boars (AY574047, 1214 bp), Indian wild boars (OM162160, 1253 bp), Duroc pig (FJ236997, 1505 bp), and Ha Lang pig (KY800118, 1285 bp). This greater size variation is attributed to the presence of multiple tandem repeats and variations in their copy numbers [35]. Additionally, one mutation at site 405 was only found in the Vietnamese Central Highland wild boar, Ha Lang pig, and Dong Khe pig, while an 11 bp duplication (TAAAACACTTA) was notably identified in the two former pigs, suggesting potentially distinctive features of pigs from Vietnam. Despite the difference in tandem repeat copy numbers, the Vietnamese Central Highland wild boar and Vietnamese wild boar exhibited only three nucleotide variations. These nucleotide variations may possibly be influenced by natural selection and genetic drift, whereas tandem repeats are potentially involved in regulating the mitochondrial genome replication process [36,37].

As observed in vertebrates, the rRNA genes are separated by the *trnV* gene [38]. The A + T content of both rRNA genes matches well with earlier findings about the mitogenome of *Sus scrofa*, showing a conserved nature of mitochondrial genomes [18].

Intergenic regions may contain functional elements that play an essential role in regulating gene expression [39]. These intergenic spacers are generally found in other *Sus scrofa* individuals, although their length may slightly vary.

Despite having similar nucleotide variations to other pig breeds, the Vietnamese Central Highland *Sus scrofa* exhibited several polymorphisms unique to Vietnamese pigs, indicating distinct genetic variation between the Vietnamese Central Highland wild boar and other pig breeds’ taxa. Notably, the mismatched base pairs identified in the tRNA genes may potentially influence the tRNA function, possibly reflecting how the wild boar adapts to its particular living habitat.

The *cytochrome b* gene has been widely used in phylogenetic and evolutionary studies of many mammalian species due to its high variability [40]. Major pig haplotypes can be classified into region-specific groups based on single-nucleotide polymorphisms at four positions 15,036 (T/C), 15,038 (G/A), 15,041 (C/T), and 15,045 (G/A) [12]. It should be noted that the positions are putative with respect to the reference sequence used. The E1 (TGCG) and E2 (TGTG) haplotypes were of European origin, whereas the A1 (CATA), A2 (CATG), and A3 (TATG) haplotypes were Asian [17,41,42]. The analysis of eight Vietnamese boars revealed that the Vietnamese Central Highland wild boar characterized in this study exhibited A1 haplotypes. While one wild boar (MZ574575.1), also from the Central Highlands, displayed A3 haplotypes, A2 haplotypes were observed in all the remaining boars, indicating the presence of all three Asian lineages within Vietnamese wild boars. The reference mitogenome displayed E1 haplotypes. Notably, a substitution at position 16,379 was found in all Vietnamese boars, thus suggesting a potential marker to differentiate Asian from European pigs.

The phylogenetic tree revealed a clear distinction between Asian and European pigs. Additionally, the tree demonstrated mutual genetic contributions between Asian and European pig breeds, consistent with evidence of introgression reported by Giuffra [41]. The wild boars analyzed in this study do not form a separate clade but are scattered among domestic pig breeds, which could be attributed to the long history of domestication, feralization, introgressive hybridization, and pig breeding [43,44,45,46,47,48].

Phylogenetic analysis showed the close genetic relationship between Vietnamese wild boars and domestic pig breeds. Interestingly, the Vietnamese Central Highland wild boar is genetically close to Vietnamese northern pigs (Ha Lang pig and Dong Khe pig), despite the regional difference. While the Vietnamese wild boar represents a close relationship to Chinese wild boars (Hainan and Yunnan wild boars), the Vietnamese Central Highland wild boar is relatively distant from other wild boars. The two Vietnamese wild boars were positioned distantly on the phylogenetic tree, possibly reflecting their origin from genetically distinct populations. Additionally, the tree indicated that the Vietnamese Central Highland wild boar was closely related to the Xiang and Mashen pigs. This result contradicts the findings of Lan and Shi, who reported a high genetic divergence between Vietnamese wild boars and Chinese domestic pigs [49].

## 5. Conclusions

The complete mitochondrial genome of the Vietnamese Central Highland *Sus scrofa* presents characteristics typically observed in other vertebrates. While this characterization serves as an initial molecular analysis of the Vietnamese wild boar, an in-depth investigation is required to further understand its genetic variation and evolutionary history. Combining genetic data with ecological and demographic data may hold great importance for effectively preserving the gene integrity of natural wild boar populations.

This study has several limitations. First, the focus on the mitochondrial genome solely reflects maternal inheritance, excluding the influence of paternal genetic contributions and recombination events, which requires more complex and extensive genomic analyses. Second, the data reported were limited to wild boars from the Central Highlands region and thus were insufficient to assess the genetic diversity across other wild boar populations throughout Vietnam. Notably, phylogenetic analysis demonstrated high genetic variations between the two Vietnamese wild boar individuals, highlighting the need for future research to evaluate the origin and population structure of Vietnamese wild boars.

Given the loss of unique wild boar traits caused by widespread hybridization, this study provides comprehensive insights into the genetic characteristics of native wild boars in the Central Highlands of Vietnam, serving as the basis for future research in conservation, evolutionary biology, and the use of wild boar genetic resources in breeding programs.

## Figures and Tables

**Figure 1 animals-15-02029-f001:**
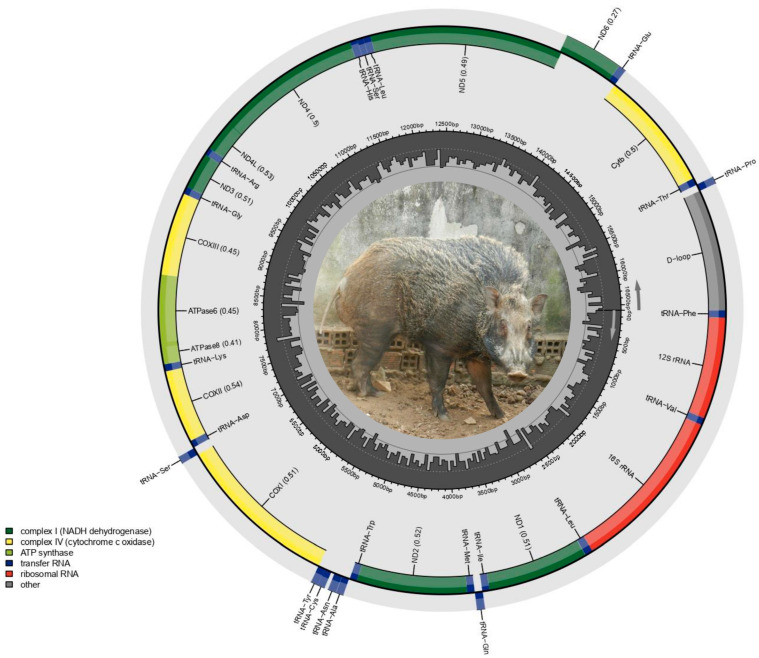
The genetic map of the complete mitochondrial genome of the Vietnamese Central Highland *Sus scrofa*. The circular mitochondrial genome is depicted, and each gene type is illustrated by a specific color. The control region is shown in gray, the 16 S (large rRNA) and 12 S (small rRNA) genes are shown in red, the 22 tRNA genes are colored in dark blue, and the 13 PCGs are in green and yellow. Genes encoded on the H strand are displayed on the outer circle, while those on the L strand are positioned on the inner circle.

**Figure 2 animals-15-02029-f002:**
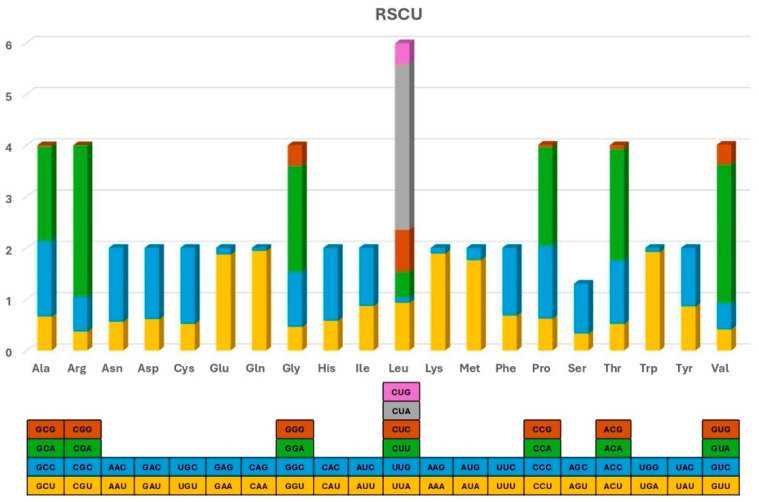
The relative synonymous codon usage (RSCU) of the protein-coding genes in the mitogenome of *Sus scrofa*. Codons that specify the amino acids are shown on the *X*-axis.

**Figure 3 animals-15-02029-f003:**
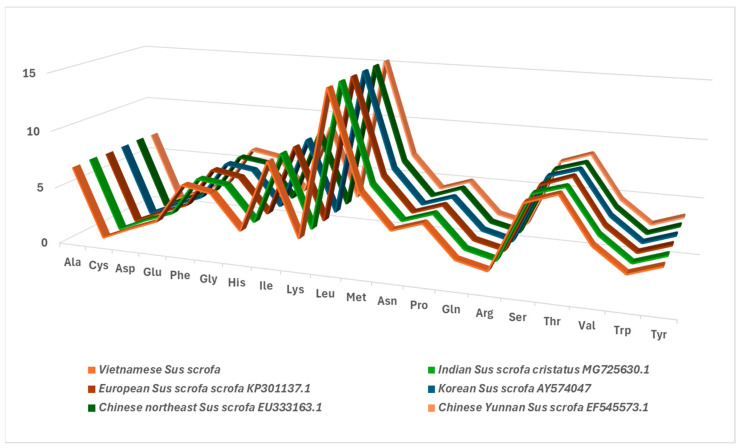
The distribution of amino acids among 6 wild boars from different geographic regions.

**Figure 4 animals-15-02029-f004:**
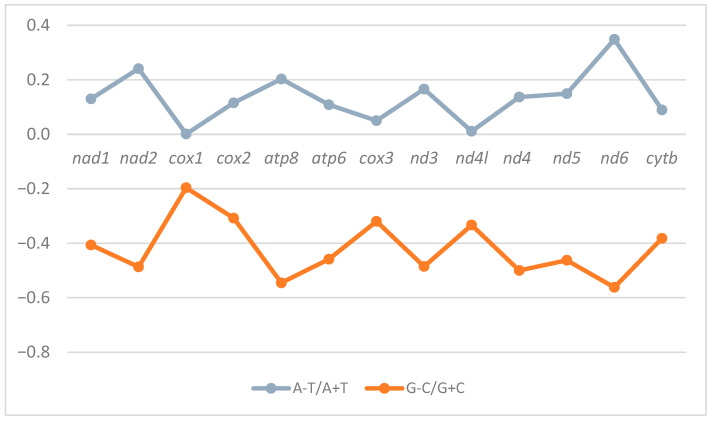
AT and GC skews of 13 protein-coding genes in the *Sus scrofa* mitogenome.

**Figure 5 animals-15-02029-f005:**
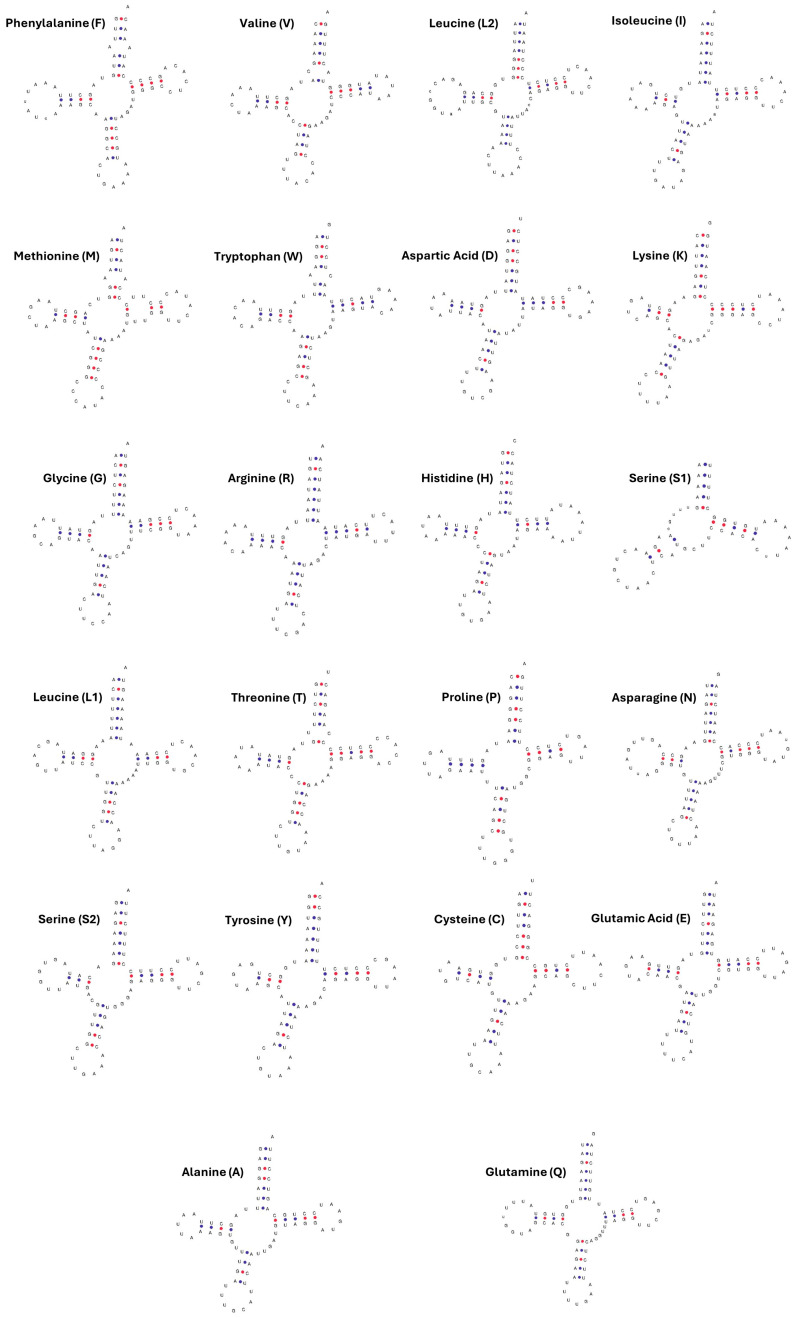
The predicted secondary structure of 22 tRNA genes in the complete mitochondrial genome of Vietnamese Central Highland *Sus scrofa*. GC and AU connections are colored in red and blue, respectively.

**Figure 6 animals-15-02029-f006:**
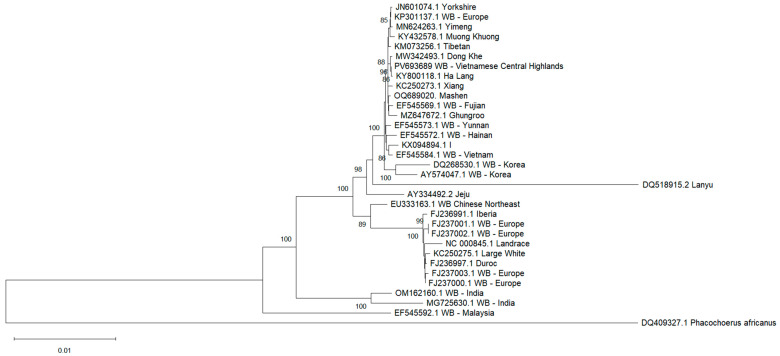
The phylogenetic tree from whole mitogenome sequences using the neighbor-joining method with 1000 bootstrap replicates. Bootstrap values are indicated on the branches of the tree. Values under 70 are not shown. WB—wild boar.

**Table 1 animals-15-02029-t001:** The gene organization of the complete mitochondrial genome of the Vietnamese Central Highland *Sus scrofa*.

Name	Start	Stop	Strand	Length	Intergenic Nucleotides
*trnF(gaa)*	345	414	+	70	0
*rrnS*	415	1376	+	962	0
*trnV(tac)*	1376	1443	+	68	−1
*rrnL*	1442	3013	+	1572	−2
*trnL2(taa)*	3014	3088	+	75	0
*nad1*	3091	4045	+	955	2
*trnI(gat)*	4046	4114	+	1042	0
*trnQ(ttg)*	4112	4184	−	73	−3
*trnM(cat)*	4186	4255	+	70	1
*nad2*	4256	5297	+	1042	0
*trnW(tca)*	5298	5365	+	68	0
*trnA(tgc)*	5372	5439	−	68	6
*trnN(gtt)*	5441	5515	−	75	1
*trnC(gca)*	5548	5613	−	66	32
*trnY(gta)*	5613	5678	−	66	−1
*cox1*	5680	7224	+	1545	1
*trnS2(tga)*	7228	7296	−	69	3
*trnD(gtc)*	7304	7371	+	68	7
*cox2*	7372	8059	+	688	0
*trnK(ttt)*	8060	8126	+	67	0
*atp8*	8128	8331	+	204	1
*atp6*	8289	8969	+	681	−43
*cox3*	8969	9752	+	784	−1
*trnG(tcc)*	9753	9821	+	69	0
*nad3*	9822	10,168	+	347	0
*trnR(tcg)*	10,169	10,237	+	69	0
*nad4l*	10,238	10,534	+	297	0
*nad4*	10,528	11,905	+	1378	−7
*trnH(gtg)*	11,906	11,974	+	69	0
*trnS1(gct)*	11,975	12,033	+	59	0
*trnL1(tag)*	12,034	12,103	+	70	0
*nad5*	12,104	13,924	+	1821	0
*nad6*	13,908	14,435	−	528	−17
*trnE(ttc)*	14,436	14,504	−	69	0
*cob*	14,509	15,648	+	1140	4
*trnT(tgt)*	15,649	15,716	+	68	0
*trnP(tgg)*	15,716	15,780	−	65	−1

**Table 2 animals-15-02029-t002:** Nucleotide composition indices in specific regions of 17 pig mitogenomes.

	Accession Number	Whole		Protein-Coding Genes (PCGs)		Large Ribosomal RNA (rrnL)		Small Ribosomal RNA (rrnS)	
		Length	%AT	Length	%AT	Length	%AT	Length	%AT
Vietnamese Central Highland *Sus scrofa*	PV693689	16,581	60.6	11,342	60.3	1572	62.7	962	59.4
Indian *Sus scrofa cristatus*	MG725630.1	16,738	60.46	11,348	60.4	1570	62.4	958	59.2
European *Sus scrofa scrofa*	KP301137.1	16,770	60.41	11,342	60.4	1570	62.8	960	59.5
Korean *Sus scrofa*	AY574047.1	16,651	60.57	11,345	60.4	1570	62.7	962	59.6
Chinese northeast *Sus scrofa*	EU333163.1	16,581	60.60	11,341	60.3	1569	62.6	962	59.5
Chinese Yunnan *Sus scrofa*	EF545573.1	16,620	60.48	11,341	60.3	1570	62.7	961	59.4
*Sus scrofa domestica*	AP003428.1	16,770	60.33	11,354	60.2	1571	62.6	962	59.5
*Sus scrofa* breed Yorkshire	JN601074.1	16,770	60.41	11,354	60.3	1571	62.8	962	59.4
*Sus scrofa* breed Tibetan	KM073256.1	16,710	60.47	11,341	60.3	1570	62.8	961	59.4
*Sus scrofa* breed Iberian	FJ236991.1	16,941	60.15	11,369	60.3	1571	62.6	962	59.5
*Sus scrofa* breed Lanyu	DQ518915.2	16,747	60.44	11,345	60.3	1563	62.8	956	58.9
*Sus scrofa* breed Large White	KC250275.1	16,610	60.55	11,341	60.3	1570	62.5	960	59.5
*Sus scrofa* breed Dong Khe	MW342493.1	16,708	60.46	11,342	60.3	1570	62.7	962	59.4
*Sus scrofa* breed Ha Lang	KY800118.1	16,717	60.47	11,338	60.3	1572	62.7	962	59.5
*Sus scrofa* breed I	KX094894.1	16,724	60.42	11,337	60.3	1570	62.7	960	59.5
*Sus verrucosus*	NC_023536.1	16,479	60.98	11,335	60.5	1570	62.6	959	60
*Phacochoerus africanus*	DQ409327.1	16,719	60.75	11,361	60.6	1572	63.2	963	59.9

**Table 3 animals-15-02029-t003:** The AT and GC skews in the protein-coding genes of 17 pig mitogenomes.

	Accession Number	T (U)	C	A	G	Total	AT Skew	GC Skew
Vietnamese Central Highland *Sus scrofa*	PV693689	26.3	27.8	34.0	11.9	11,342	0.127156	−0.40089
Indian *Sus scrofa cristatus*	MG725630.1	26.4	27.7	34.0	11.9	11,348	0.12542	−0.39942
European *Sus scrofa scrofa*	KP301137.1	26.3	27.8	34.0	11.9	11,342	0.127246	−0.40093
Korean *Sus scrofa*	AY574047.1	26.3	27.7	34.0	12.0	11,345	0.127793	−0.39618
Chinese northeast *Sus scrofa*	EU333163.1	26.3	27.8	34.0	11.9	11,341	0.128603	−0.40124
Chinese Yunnan *Sus scrofa*	EF545573.1	26.3	27.8	34.0	11.9	11,341	0.126901	−0.40013
*Sus scrofa domestica*	AP003428.1	26.2	27.8	34.0	11.9	11,354	0.12957	−0.40035
*Sus scrofa* breed Yorkshire	JN601074.1	26.3	27.7	34.0	11.9	11,354	0.127262	−0.39938
*Sus scrofa* breed Tibetan	KM073256.1	26.3	27.8	34.0	11.9	11,341	0.126991	−0.40062
*Sus scrofa* breed Iberian	FJ236991.1	26.3	27.8	34.0	11.9	11,369	0.128228	−0.40142
*Sus scrofa* breed Lanyu	DQ518915.2	27.4	26.9	32.9	12.8	11,345	0.091732	−0.35586
*Sus scrofa* breed Large White	KC250275.1	26.2	27.9	34.0	11.9	11,341	0.12864	−0.40151
*Sus scrofa* breed Dong Khe	MW342493.1	26.3	27.8	34.0	11.9	11,342	0.127448	−0.40133
*Sus scrofa* breed Ha Lang	KY800118.1	26.3	27.8	34.0	11.9	11,338	0.127358	−0.4012
*Sus scrofa* breed I	KX094894.1	26.3	27.8	34.0	11.9	11,337	0.12714	−0.40027
*Sus verrucosus*	NC_023536.1	26.5	27.6	34.0	11.8	11,335	0.124454	−0.4004
*Phacochoerus africanus*	DQ409327.1	26.6	27.5	34.0	12.0	11,361	0.122802	−0.39375

**Table 4 animals-15-02029-t004:** The mismatched base pairs from mitochondrial tRNA genes of *Sus scrofa*. AA—amino acid acceptor, T-arm—pseudouridine, AC—anticodon.

tRNA	Mismatched Base Pairs	Stem	Frequency
Methionine CAT	U-U A-A	T-arm AA	2 1
Phenylalanine GAA	A-A	AA	1
Serine GCT	A-A	AC	1
Threonine TGT	U-C	AA	1
Tryptophan TCA	A-C	AA	1
Valine TAC	C-A	AA	1

## Data Availability

Data are contained within the article and Appendix A.

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
