# Peer review of "Characterization of the Mitochondrial Genome of the Vietnamese Central Highland Wild Boar (Sus scrofa)"

_animals, 2025, doi:10.3390/ani15142029_

Round 1
Reviewer 1 Report
Comments and Suggestions for Authors
The manuscript entitled “Characterization of the mitochondrial genome of the Vietnamese Central Highland wild boar (Sus scrofa)” presents a relevant contribution to the field of animal genetics, especially in the context of conservation and a better understanding of the genetic diversity of wild boars in Vietnam. The study provides a comprehensive description of the mitochondrial genome, combining modern bioinformatics tools with comparative and phylogenetic analyses. The approach is scientifically sound and addresses a significant gap in the literature. However, before the manuscript can be considered for publication, the authors should carefully address some issues and necessary improvements, as described below:
Abstract
Line 44: Sus scrofa should be written in italics.
Line 49: Avoid using keywords that already appear in the title, such as “Sus scrofa” and "wild boar".
Introduction
The introduction needs to be better developed regarding genetic data on Sus scrofa, including genetic variability among populations, population structure, etc. These elements would help emphasize and justify the relevance of the study.
Lines 63–67: What is the reference for this information?
Material and Methods
Lines 129, 131, 134, and 137: Sus scrofa should be written in italics.
Lines 140–149: The phylogenetic tree should be constructed using all 13 protein-coding genes. There is no rationale for conducting the analysis using only Cytochrome b, especially given that the genome has been described and assembled.
Results
Throughout the Results section, there are several instances where the authors begin interpreting and discussing the findings. However, such interpretations should be reserved for the Discussion section, which currently lacks depth and critical analysis. Therefore, I strongly recommend a careful revision of the Results section to separate descriptive content from interpretation. Specific passages that would be more appropriate in the Discussion include: Lines 203–207; 214–220; 227–234; 253–257; 274–277; 289–303; and 312–319. These segments contain content that would greatly enrich the Discussion and should be relocated accordingly.
Lines 162, 183, 192, and 210: Sus scrofa should be italicized.
Line 184: Figure 3 is not visually appropriate; please format it to match the style of the other figures.
Lines 235–237: Table 4 should be moved to the supplementary materials, as it is not a critical result for the manuscript.
Lines 304–306: Table 6 should also be placed in the supplementary materials, as it is not essential to the core findings.
Lines 307–319: I understand the relevance of using Cytochrome b for analysis; however, since the manuscript describes a complete mitochondrial genome, the analyses should be based on the 13 protein-coding genes. These should be compared with other available genomes from the species and related taxa. This section, therefore, needs to be revised accordingly.
Discussion
Lines 327–328: This statement is incorrect. There are numerous other studies that have conducted molecular characterizations of the mitochondrial genome in this species. While your study includes some additional analyses, the claim of being the first such study should be removed.
Lines 338–360: This portion of the discussion, which represents almost two-thirds of the section, is based on the Cytochrome b phylogeny. In my opinion, this is not appropriate for a study focused on mitochondrial genome characterization. The phylogenetic analysis should instead be performed using the 13 protein-coding genes. Additionally, the discussion should be revised and expanded to explore other aspects, such as differences in control region length, genetic conservatism across populations, and other relevant features.
Conclusions
I suggest including comments on the study’s limitations and its potential applications.
Author Response
Dear Prof. Dr. Clive J. C. Phillips, Editorial Board, and Reviewers,
We are very grateful to the Editor for your consideration of our manuscript. We would like to thank the Reviewers for your careful and thorough reading of the manuscript and for the thoughtful comments and constructive suggestions, which help improve the quality of this manuscript. Each comment has been carefully considered point by point and responded. Responses to the reviewers and changes in the revised manuscript are as follows.
Reviewer 1:
Abstract
Comment 1: Sus scrofa should be written in italics (Line 44, 129, 131, 134, and 137)
Response 1: Thank you for your comment. The species names were rewritten in italic
Comment 2: Avoid using keywords that already appear in the title, such as “Sus scrofa” and "wild boar" (Line 49).
Response 2: Keywords already mentioned in the title were removed.
Introduction
Comment 3: The introduction needs to be better developed regarding genetic data on Sus scrofa, including genetic variability among populations, population structure, etc. These elements would help emphasize and justify the relevance of the study.
Response 3: Thank you for your valuable suggestion. We agree that information on genetic variability, population structure, and broader population-level data of Sus scrofa would help emphasize the relevance of our work. However, the primary aim of this study is to characterize the complete mitochondrial genome of a Vietnamese Central Highland wild boar individual, which is rarely encountered in nature. Given the scope of the current study, we have not revised the Introduction but will consider incorporating broader population-level context in subsequent studies.
Comment 4: What is the reference for this information? (Line 63 – 67)
Response 4: The reference was added.
Materials and Methods
Comment 5: The phylogenetic tree should be constructed using all 13 protein-coding genes. There is no rationale for conducting the analysis using only Cytochrome b, especially given that the genome has been described and assembled.
Response 5: The phylogenetic was reconstructed using whole mitogenomes.
Results
Comment 6: Throughout the Results section, there are several instances where the authors begin interpreting and discussing the findings. However, such interpretations should be reserved for the Discussion section, which currently lacks depth and critical analysis. Therefore, I strongly recommend a careful revision of the Results section to separate descriptive content from interpretation. Specific passages that would be more appropriate in the Discussion include: Lines 203–207; 214–220; 227–234; 253–257; 274–277; 289–303; and 312–319. These segments contain content that would greatly enrich the Discussion and should be relocated accordingly.
Response 6: We relocated the interpretative content to the Discussion, adding more explanations and analysis to emphasize the findings.
Comment 7: Figure 3 is not visually appropriate; please format it to match the style of the other figures (Line 184)
Response 7: Thank you for your feedback! The analysis of amino acid profile is an essential feature that should be addressed in mitogenome characterization. This figure was generated using the same visual style and method established in prior study (please refer to doi:10.3390/cimb46090592 ; doi:s41598-018-20946-5), including one that was previously published under MDPI. For that reason, we believe the current format is appropriate and have retained the original figure without any modification. We hope for your understanding in this matter.
Comment 8: Table 4 and Table 6 should be moved to the supplementary materials, as it is not a critical result for the manuscript (Line 235-237; line 304-306)
Response 8: The tables were moved to the Supplementary materials.
Comment 9: I understand the relevance of using Cytochrome b for analysis; however, since the manuscript describes a complete mitochondrial genome, the analyses should be based on the 13 protein-coding genes. These should be compared with other available genomes from the species and related taxa. This section, therefore, needs to be revised accordingly. (Line 307-319)
Response 9: The phylogenetic tree construction based on cytochrome b gene was removed. However, we retained the analysis of partial cytochrome b (but limited to Vietnamese wild boars only) to assess their genetic origin using 4 specific SNPs. This investigation contributes more valuable insights into the genetic diversity of Vietnamese wild boars.
Discussion
Comment 10: This statement is incorrect. There are numerous other studies that have conducted molecular characterizations of the mitochondrial genome in this species. While your study includes some additional analyses, the claim of being the first such study should be removed. (Line 327-328)
Review 10: The statement was edited according to the reviewer’s comment
Comment 11: This portion of the discussion, which represents almost two-thirds of the section, is based on the Cytochrome b phylogeny. In my opinion, this is not appropriate for a study focused on mitochondrial genome characterization. The phylogenetic analysis should instead be performed using the 13 protein-coding genes. Additionally, the discussion should be revised and expanded to explore other aspects, such as differences in control region length, genetic conservatism across populations, and other relevant features. (Lines 338–360)
Response 11: The phylogenetic tree construction based on cytochrome b gene was removed. However, we retained the analysis of partial cytochrome b (but limited to Vietnamese wild boars only) to assess their genetic origin using 4 specific SNPs. This investigation contributes more valuable insights into the genetic diversity of Vietnamese wild boars. We relocated the interpretative content to the Discussion, adding more explanations and analysis to emphasize the findings.
Conclusions
Comment 12: I suggest including comments on the study’s limitations and its potential applications.
Response 12: Study’s limitations and potential applications were added.
We hope that the revision of our manuscript could meet your requirements.
Best regards
Reviewer 2 Report
Comments and Suggestions for Authors
The article is very interesting. It evaluates the mitogenome of a wild boar clearly and rigorously. However, I have a few suggestions about the document, which I will list according to the line numbers.
L94: It looks like an error in 4ºC.
L109: Were all three equipment sets used? What did this depend on? Please clarify.
L117: replace the word ‘purified’ with a more appropriate one
L160: I recommend specifying how many genes are on the H and L strands.
L198: Rewrite this sentence to make it closer.
L202-203: Is there only one explanation? Could there be another? Could it be due to sequencing problems?
L261: The coloring does not look satisfactory (Figure 5)
L326-360 (Discussion): The discussion was already taking place in the results section. I think it is possible to have both together (although it is not recommended).
Author Response
Dear Prof. Dr. Clive J. C. Phillips, Editorial Board, and Reviewers,
We are very grateful to the Editor for your consideration of our manuscript. We would like to thank the Reviewers for your careful and thorough reading of the manuscript and for the thoughtful comments and constructive suggestions, which help improve the quality of this manuscript. Each comment has been carefully considered point by point and responded. Responses to the reviewers and changes in the revised manuscript are as follows.
Reviewer 2
Comment 1: It looks like an error in 4ºC (Line 94)
Response 1: The error was corrected.
Comment 2: Were all three equipment sets used? What did this depend on? Please clarify. (Line 109)
Response 2: The sequencing system is specified.
Comment 3: Replace the word ‘purified’ with a more appropriate one (Line 117).
Response 3. The word “purified” is replaced with “pre-processed”.
Comment 4: I recommend specifying how many genes are on the H and L strands. (Line 160)
Response 4: The number of genes located on each strand was specified.
Comment 5: Rewrite this sentence to make it closer. (Line 198)
Response 5: To make it clearer, the sentence was rewritten.
Comment 6: Is there only one explanation? Could there be another? Could it be due to sequencing problems? (Line 202-203)
Response 6: Thank you for your comment! Incomplete termination codon in PCGs is a biological feature observed in mitochondrial genomes. This mechanism has been well-documented in mitochondrial gene expression of vertebrates, including mammals. The regions containing the incomplete stop codons were supported by high read depth and clean base quality scores, thus reducing the likelihood of sequencing errors. The same incomplete stop codons are present in closely related mitochondrial genomes, indicating that they are biologically conserved.
Comment 7: The coloring does not look satisfactory (Figure 5) (Line 261)
Response 7: We appreciate the reviewer’s comment regarding the coloring of this figure. Unfortunately, the tRNA structures were generated via a web-based platform that does not allow manual customization of the designs. As such, we are unable to modify the figure to meet specific aesthetic preferences. We hope for your understanding in this matter.
Comment 8: The discussion was already taking place in the results section. I think it is possible to have both together (although it is not recommended). (Line 326-360)
Response 8: We relocated the interpretative content to the Discussion, adding more explanations and analysis to emphasize the findings.
We hope that the revision of our manuscript could meet your requirement.
Best regards
Round 2
Reviewer 1 Report
Comments and Suggestions for Authors
Although I insist that it is important to improve the introduction, I will not insist on the issue and leave it to the editor's discretion. Otherwise, the manuscript is suitable for publication.